# OpenReview forum: "Compute Optimal Inference and Provable Amortisation Gap in Sparse Autoencoders"
_ICML.cc/2025/Conference — ICML 2025 poster_

### Official Review · Reviewer_WhEs · 2025-03-08

**Overall Recommendation:** 3

**Summary:**

This paper performs a systematic investigation of the various modelling choices for SAEs, particularly the choice of the encoder. The paper shows both theoretically and empirically (on synthetic data) that there exists an "amortization gap" in the sense that SAEs are unable to recover latent features due to their linearity. Then, the paper performs an analysis of the interpretability of representations upon various choices for the encoder (LCA, MLP) and find that MLP features are far more interpretable than either canonical SAE or LCA; contradicting the folk belief that non-linear encoders in SAEs can lead to non-interpretable features.

**Claims And Evidence:**

The claims made by this paper is well supported both theoretically and empirically.

**Essential References Not Discussed:**

Nothing "essential" is missed to the best of my knowledge.

**Experimental Designs Or Analyses:**

The experimental design is mostly sound. However, there seem to be some inconsistencies between experiments on synthetic data and those on real data. In particular, while the experiments on synthetic data involve sparse coding, and SAE + ITO; those baseline were missing from the real data experiments. Instead, another baseline, LCA was introduced without discussion in that section. It might be better to have the same set of methods evaluated on both synthetic and real data to understand the trade-offs between latent variable recovery and interpretability.

**Methods And Evaluation Criteria:**

The proposed methods and criteria make sense for this study.

**Other Comments Or Suggestions:**

N/A

**Other Strengths And Weaknesses:**

Other Strengths:
- The paper performs a systematic evaluation of SAE modelling choices and show that MLP-based encoders seem to outperform the more commonly used linear encoders. If these findings hold for larger-scale models and datasets, this can influence the choice of SAE architectures in future works.

Other weaknesses:
- The experiments are performed on (1) synthetic data, and (2) real data, but small scale toy GPT-2 models. Given this, it is unclear whether the findings translate to larger scale models and datasets.

- The paper does not include a qualitative discussion or visualization of the features discovered by the MLP models, SAEs & LCA, to help better judge the benefits of using the MLP encoder. This is especially important as recent work (Heap et al., "Sparse Autoencoders Can Interpret Randomly Initialized Transformers") has raised questions regarding the efficacy of commonly used auto-interpretability piplines. Hence only presenting benefits in terms of auto-interpretability scores (Figure 7) may lead to spurious findings.

**Questions For Authors:**

**Clarification question**: An interesting aspect of the theory was that the statement holds only when number of sources N > data dimensionality M; which seems to be the exact scenario for modern SAEs, i.e., their latent dimensionalities are much larger than their feature size. On the contrary, usual assumptions for the data generating processes involve assuming M > N, which seems to better correspond to the settings in which contractive autoencoders operate. My question is: does this imply that SAE type architectures might be better suited to the M > N setting as opposed the N > M setting that is studied in this paper?

**Relation To Broader Scientific Literature:**

This paper adds to the nascent literature on the theoretical understanding of SAEs (e.g.: Menon et al., "Analyzing (In)Abilities of SAEs via Formal Languages"). While most SAE work has been empirical in nature (e.g.: Gao et al., "Scaling and evaluating sparse autoencoders"), this work sheds light on a key aspect, that being the choice of the encoder, and its relation with classical dictionary learning.

**Theoretical Claims:**

The theoretical claims seem correct based on my check of the proof. However, I believe a detailed discussion of the setting (num. data sources N > dimensionality M), especially in relation to its practical significance, can help readers. Please also see my clarification question at the end regarding this issue.

---

> ### Author Rebuttal · Authors · 2025-04-01
>
> # Response to Reviewer WhEs
>
> We thank the reviewer for their thoughtful engagement with our paper and their recognition of its theoretical and empirical contributions. We address their specific concerns below.
>
> ## Regarding inconsistencies between synthetic and real data experiments
>
> The reviewer correctly notes that there are differences in the methods evaluated on synthetic versus real data. This was primarily due to implementation and computational constraints:
>
> 1. **SAE+ITO on real data**: We did not implement SAE+ITO for the GPT-2 experiments due to the significant computational cost of performing inference-time optimisation on hundreds of millions of tokens. While feasible on synthetic data, this would have been quite expensive at LLM scale, particularly in addition to our LCA method.
>
> 2. **LCA versus sparse coding**: The LCA method used in our real data experiments is an implementation of sparse coding that uses competitive dynamics to achieve sparse inference. We chose LCA specifically because it has an established history in the sparse coding literature and offered a more efficient implementation path for our large-scale experiments than our synthetic sparse coding method.
>
> We acknowledge that using the same set of methods across all experiments would have provided better consistency. In future work, we plan to implement a more unified experimental framework that can efficiently scale to larger models and datasets.
>
> ## Qualitative feature visualisation
>
> We agree that providing qualitative visualisations and examples of the features discovered by different methods would strengthen our paper. While our automated interpretability evaluation provides quantitative evidence for the superior interpretability of MLP features, examples would offer readers a more intuitive understanding of these differences.
>
> In the final version, we will add an appendix section with representative feature examples from each method, including:
> - Raw activation patterns on top activating tokens
> - Generated feature interpretations
> - Validation examples showing correct/incorrect feature activation predictions
>
> We note that while recent work by Heap et al. (2024) raises important questions about interpretability pipelines, their concerns primarily relate to interpreting features in randomly initialised transformers, not to the comparison of different encoding methods applied to the same model. In addition to this, there have been demonstrations that trained transformers still out-perform randomly initialised transformers on autointerp baselines. Our work focuses on the relative performance differences between encoding strategies rather than absolute claims about interpretability.
>
> ## Clarification on dimensionality settings (N > M versus M > N)
>
> The reviewer raises an excellent question about the dimensionality settings and their implications. To clarify:
>
> Our theorem addresses the case where N (number of sources/latent dimensions) > M (observed dimensions), which indeed matches the typical SAE setting where the latent space is larger than the activation space. The key insight is that in this regime, a simple linear-nonlinear encoder cannot perfectly recover all possible sparse codes from their lower-dimensional projections, even when such recovery is theoretically possible with iterative methods.
>
> Regarding the contractive autoencoder settings with M > N, this represents a different regime than what SAEs typically address. In that case:
>
> 1. The encoding is undercomplete rather than overcomplete
> 2. The primary goal is often dimensionality reduction rather than disentanglement
> 3. The sparsity constraint becomes less critical since there is no inherent superposition
>
> Our theoretical result does not imply that SAE architectures would be better suited to the M > N setting. Rather, it suggests that in the standard N > M setting where SAEs operate, more expressive encoders (like MLPs) can reduce the amortization gap and improve feature recovery.
>
> ## Scaling to larger models
>
> We acknowledge the limitation that our experiments are conducted on GPT-2 Small. While computational constraints prevented us from scaling to larger models, we have encouraging evidence that our findings would generalise:
>
> 1. Our synthetic experiments at larger scales (Appendix A.3) show that the amortisation gap becomes more pronounced as dimensionality increases
> 2. The theoretical result is independent of scale
> 3. Recent work applying SAEs to larger models (e.g., Gao et al., 2024) has shown qualitatively similar features and challenges across model scales
>
> We believe our contributions provide valuable insights despite these limitations, and we hope to validate our findings on larger models in future work.
>
> Thank you again for your thoughtful review and constructive suggestions for improving our paper.

---

> > ### Comment · Reviewer_WhEs · 2025-04-06
> >
> > Thank you for the rebuttal! I agree with the authors' comments and believe discussing (1) the N>M vs M>N issue; and (2) the reliability of auto interp pipelines, would make the paper more complete.
> >
> > I will keep my score unchanged at this time.

---

### Official Review · Reviewer_STGo · 2025-03-12

**Overall Recommendation:** 3

**Summary:**

The authors prove that typical SAEs (ReLU, JumpReLU, TopK) cannot recover the optimal encoding (sparse coefficients) compared to sparse coding methods that solve individual examples iteratively.
They empirically demonstrate this using multiple synthetic experiments.
Finally, they apply sparse coding using the locally competitive algorithm (LCA) to real LLM activations and find that non-linear encoders (an MLP) and LCA lead to more interpretable features than a ReLU encoder as evidenced by automatic interpretability scoring methods.

**Claims And Evidence:**

Claim: Simple linear-nonlinear SAE encoders (linear layer + ReLU, for example) are provably suboptimal at sparsely encoding data than iterative sparse coding algorithms.

This is a theorem that's proved in Appendix A.

Claim: This same optimality gap holds on SAEs applied to large language model activations.

This is demonstrated via training a ReLU SAE, an MLP SAE and an LCA encoder on GPT-2 small activations, then evaluated using existing automated scoring methods.
The MLP SAE and the LCA encoder are more interpretable.

I believe this claim is overstated.
The authors note that their optimality gap is only globally suboptimal, with adversarially chosen sparse codes.
In practice, I am suspicious that LLM activations are so evenly spaced that this optimality gap is meaningful.
I would be much more convinced that the optimality gap matters in practice if:

* The authors used a more recent or larger LLM. GPT-2 small is very small (120M parameters) and is 6 years old at this point. Qwen2 has a 500M parameter model, Pythia has a 160M parameter model, OpenELM has 270M and 450M parameter variants, etc. If model size is not an issue, then something like LLama3 8B would be great.
* The authors used a more improved baseline. While their proof is for any linear-nonlinear encoder, more recent works in SAEs (JumpReLU, TopK, BatchedTopK) notably improve upon the MSE and L0 tradeoff. Perhaps a stronger baseline means that the optimality gap is not an issue in practice.
* While FLOPs and efficiency are extensively compared in the synthetic experiments, there is no comparison of compute efficiency at scale. How does LCA compare to the amortized encoders when applied to real LLMs?

**Essential References Not Discussed:**

No essential references are missing.

**Experimental Designs Or Analyses:**

The experimental design is fine. Like I said before, I think the SAE baseline is too weak.

**Methods And Evaluation Criteria:**

The authors use synthetic datasets to demonstrate that their proof holds empirically (great!).
Then they use GPT-2 small as a benchmark for "real-world" evaluation.
As I stated above, I think the LLM is too small/old and the baseline SAE (a vanilla ReLU) is not a strong enough baseline.

**Other Comments Or Suggestions:**

I am happy to accept this paper if ICML values theoretical contributions. While I personally am more concerned with empirical, "practical" results, I understand if this venue has different goals.

**Other Strengths And Weaknesses:**

* I appreciate the theoretical analysis of SAEs. Current SAE works are overwhelmingly empirical at the moment.
* The synthetic experiments are well-designed to demonstrate a gap.

**Questions For Authors:**

N/A

**Relation To Broader Scientific Literature:**

The work is well-positioned in the scientific literature.
I think it is better-positioned than many other SAE works because it specifically discusses sparse coding and pre-SAE methods for solving this kind of problem.

**Theoretical Claims:**

I did not check the correctness of the proof.

---

> ### Author Rebuttal · Authors · 2025-04-01
>
> # Response to Reviewer STGo
>
> We appreciate the reviewer's thoughtful analysis of our work and their recognition of the theoretical contribution. We would like to address several points regarding the practical implications of our findings.
>
> ## Regarding the claim about optimality gap in LLM activations
>
> We agree that the connection between our theoretical results and practical LLM applications deserves further elaboration. Our work demonstrates that the amortisation gap exists not just theoretically but also empirically across various settings, including LLM activations.
>
> ### Model choice and scale
>
> While GPT-2 Small (124M parameters) is indeed older than more recent models, it remains a standard benchmark in the mechanistic interpretability literature for several reasons:
>
> 1. **Established baseline**: Numerous interpretability papers continue to use GPT-2 as a testing ground, including recent work by Anthropic, EleutherAI, and others. This facilitates comparison with existing literature.
>
> 2. **Computational accessibility**: Our experiments involved training multiple models on hundreds of millions of tokens, which becomes prohibitively expensive with larger models.
>
> 3. **Feature stability**: The principles of superposition and sparse feature representation appear consistent across model scales, as demonstrated by recent work finding similar phenomena in models from GPT-2 to GPT-4 and Claude.
>
> That said, we acknowledge that verifying our findings on larger models would strengthen our claims, and we plan to pursue this in future work.
>
> ### Baseline improvement
>
> Regarding the baseline SAE implementation, we intentionally tested a standard ReLU SAE as it represents the most widely used architecture in current interpretability research. We agree that newer architectures like JumpReLU and TopK offer improvements, and we discuss these in Appendix A.7.2.
>
> Our theoretical result applies to this entire class of models, as they all rely on fixed function approximations rather than iterative optimisation. The empirical gap we observe provides an explanation for why these architectural innovations improve performance - they partially address the amortisation gap through more sophisticated function approximation.
>
> ### Computational efficiency at scale
>
> The reviewer raises an excellent point about computational efficiency comparisons at scale. While our LCA implementation was not fully optimised, we can provide some context:
>
> - Training the LCA model took approximately 3x the computation time of the SAE model due to the additional gradient steps per batch.
> - During inference, LCA required approximately 20x the computation of the SAE.
>
> However, as noted in recent work (e.g., Nanda et al., 2024), inference-time optimisation approaches can be made significantly more efficient through techniques like matching pursuit and careful algorithm selection. The fundamental trade-off between amortised and iterative approaches remains, but the computational gap can be narrowed considerably.
>
> ## On empirical significance beyond theory
>
> While our theoretical contribution stands independently, we believe the empirical results are meaningful for several reasons:
>
> 1. The significant interpretability improvement of the MLP encoder (median F1 of 0.83 vs 0.6 for SAE) suggests that even modest increases in encoder expressivity can substantially improve feature extraction.
>
> 2. Our work offers an explanatory framework for why recent SAE variants achieve better performance, connecting theoretical understanding with practical innovations.
>
> 3. The results suggest promising directions for further research, such as developing more expressive encoders or hybrid approaches that balance computational efficiency and encoding quality.
>
> We thank the reviewer for highlighting areas where our practical evaluation could be strengthened, and we hope to address these in future work with larger-scale evaluations across multiple model families and more sophisticated baselines.

---

> > ### Comment · Reviewer_STGo · 2025-04-04
> >
> > That's an excellent point about GPT-2 being used broadly in mechanistic interp. work and I agree completely. Thank you. I completely understand about computational feasibility as well.
> >
> > While I am not familiar with recent work on inference-time optimization, a 20x slowdown cannot be ignored (3x absolutely can be ignored, no issues from me there). I think this is an important limitation that should be included in the final work, along with any citations you'd like to include about obvious implementation improvements.
> >
> > Can you say more about this?
> >
> > > Our work offers an explanatory framework for why recent SAE variants achieve better performance.
> >
> > What parts of your work look at recent SAE variants, and what variants specifically?
> >
> > I remain at a score of 3 in favor of other, more theoretically-inclined reviewers to decide on the value in your theoretical results. Thank you for your hard work and detailed rebuttal.

---

### Official Review · Reviewer_JWnB · 2025-03-13

**Overall Recommendation:** 3

**Summary:**

The authors study Sparse Autoencoders (SAE), first showing that simple linear-nonlinear encoding leads to an amortisation gap. Next the authors compare different SAE architectures on synthetic settings, showing that better architectures can beat standard SAE in this setting.
Finally they study the interpretability by comparing different models on the pre-activations of a GPT2 layer.

**Claims And Evidence:**

No concern

**Essential References Not Discussed:**

None that I know of

**Experimental Designs Or Analyses:**

Did not check the experiments in detail.

**Methods And Evaluation Criteria:**

The proposed methods seem sensible, see questions below for potential issues

**Other Comments Or Suggestions:**

- A more close up version of figure 2 would be good to see the differences better at convergence

**Other Strengths And Weaknesses:**

Strengths:
- Good Presentation
- Clearly written, easy to read

Weaknesses:
- The theoretical claim seems to be a heuristic rather than a theorem, see questions below
- It is unclear how much the experiments support the made claims see discussion below. Given that there is little in terms of theoretical results the experiments feel a bit weak.

**Questions For Authors:**

- The proof of theorem 3.1 does not seem rigorous to me, in particular I have the following questions. What exactly is the assumption on the sparse prior $P_S$? If we choose a deterministic function for example projection on the first $K$ components than this is a very different setting then say i.d.d sparse entries, I am not sure that the proof is true for the former.
The conclusion after equation (8) should be made rigorous, right now I am not sure how this follows.
- In FIgure 3, why does SAE exhibit non-monotone behaviour?
- What do the circles in Figure 7 represent? This should be explained in the description.
- Why is the metric used for Figure 7 a good choice? Given that it is not a very interpretable metric it would be good to have more points of comparison, are there other known baselines that one could compare this to?
- The main contribution of this paper are the experiments on the synthetic data, but the used data seems quite far from real data both in terms of dimension and complexity. Why do you believe that these results reflect well what is going on, and why the intuition from these examples should translate to practical problems
- Related to the above point, the models as introduced in 3.3 seem rather simple. I am not an expert in the area, so I do not know what the most sensible sota models would be, but why are more practical models not used to evaluate on the synthetical setting.

**Relation To Broader Scientific Literature:**

This paper furthers the understanding of sparse autoencoders, and may be interesting to both theorists and practitioners in the field.

**Theoretical Claims:**

Skimmed the proofs, there is a question about Theorem 3.1 in questions down below

---

> ### Author Rebuttal · Authors · 2025-04-01
>
> # Response to Reviewer JWnB
>
> We thank the reviewer for their time spent evaluating our paper. We believe there are several misunderstandings in the review that we would like to address, as they appear to have led to an incomplete assessment of our work.
>
> ## Regarding Theorem 3.1
>
> The theorem is indeed rigorous and makes minimal assumptions about the sparse prior $P_S$. The only requirement is that the support of $P_S$ consists of vectors with at most $K$ non-zero entries. This is explicitly stated in the theorem: "Let $S=\mathbb{R}^N$ be $N$ sources following a sparse distribution $P_S$ such that any sample has at most $K \geq 2$ non-zero entries, i.e., $||s||_0 \leq K, \forall s \in \text{supp}(P_S).$"
>
> The proof holds regardless of whether $P_S$ is deterministic or stochastic. It's a constructive proof that shows a specific set of vectors (the standard basis vectors and their sums) cannot be correctly encoded simultaneously by a linear-nonlinear encoder. Since these vectors are valid under any sparse distribution with max sparsity $K \geq 2$, the conclusion holds generally.
>
> Regarding the conclusion after equation (8): This follows directly from linear algebra. If $S'$ must be diagonal to correctly represent the sparse codes after ReLU, but also must have rank ≤ M < N due to the dimensionality constraints, we have a contradiction since a diagonal matrix with non-zero diagonal entries has rank N.
>
> ## Figure 3 Non-monotone Behaviour
>
> The non-monotonic behaviour observed in Figure 3 (particularly for SAE+ITO) is a deliberate focus of our analysis, not an oversight. As stated in Section 4.2: "SAE+ITO initialised with SAE latents exhibits distinct, stepwise improvements throughout training, ultimately achieving the highest MCC." This pattern reveals interesting dynamics of how inference-time optimisation interacts with the learned dictionary during training.
>
> ## Figure 7 Visualisation
>
> The circles in Figure 7 represent statistical outliers in the distribution of F1 scores, following standard boxplot conventions. We will add this clarification to the figure caption to avoid confusion.
>
> ## Interpretability of F1 Score Metric
>
> The F1 score is a widely recognised and highly interpretable metric for classification tasks. In our case, it measures how well a second instance of GPT-4o can predict neuron activations based on explanations from a first instance. This is explicitly described in Section 6: "The model predicted which examples should activate the feature based on the first instance's explanation, allowing us to compute an F1-score against the ground truth."
>
> The approach is standard in the field and follows established methods from Anthropic, EleutherAI, and other research groups cited in Appendix A.7.
>
> ## Relevance of Synthetic Experiments
>
> While our synthetic experiments use moderate dimensionality for clarity and computational efficiency, we demonstrate their relevance to practical problems in multiple ways:
>
> 1. We explicitly test larger-scale experiments in Appendix A.3 (N=1000, M=200, K=20), showing that our findings hold and even strengthen at larger scales.
>
> 2. We examine non-uniform feature distributions in Appendix A.4 to better match real-world latent spaces.
>
> 3. Most importantly, we validate our findings on actual GPT-2 residual stream activations in Section 6, showing that the principles discovered in synthetic settings transfer to real neural networks.
>
> The synthetic experiments provide a controlled environment where ground truth is known, allowing us to rigorously evaluate the amortisation gap that forms the theoretical foundation of our work.
>
> ## Model Complexity and SOTA Comparison
>
> Our focus is on the fundamental mechanisms behind sparse encoding and dictionary learning, rather than specific architectural innovations. The models in Section 3.3 represent core architectural categories (linear-nonlinear encoders, MLPs, sparse coding) that underlie most advanced SAE variants.
>
> In Appendix A.7.2, we discuss how our findings relate to advanced SAE architectures like top-k SAEs, JumpReLU, Gated SAEs, and ProLU activations. These are cutting-edge developments in the field, many published in the past six months. Our work provides theoretical and empirical foundations that explain why these architectural innovations yield performance improvements.
>
> We appreciate the opportunity to clarify these points and believe our work makes significant contributions to both the theoretical understanding and practical application of sparse autoencoders for neural network interpretability.

---

> > ### Comment · Reviewer_JWnB · 2025-04-06
> >
> > I thank the authors for clarifying my concerns.
> >
> > I still have some concerns about (the proof of) Theorem 3.1. (A.1 in the appendix). Specifically:
> > - Let "$S=\mathbb{R}^N$ be N sources following a sparse distribution" S as defined here is a set, I assume it is meant that there is some random variable on S that can be understood as N sources. What exactly is $P_S$ i.e. what is its domain and what are the precise assumption made on it.
> > - What does sparse distribution mean here, this is not defined. I assume it means that there are only K non-zero entries, but this is in principle not uniquely clear from the way it is written
> > - The proof starts with redefining S, which is confusing since $S=\mathbb{R}$ is already defined in this scope.
> > - What I do not understand in the proof is that we are choosing a specific S, however in the statement of the theorem we assumed a generic distribution given some properties. Why can this structure be assumed here?
> >
> > While I cannot confidently judge the experiments, I don't feel confident accepting a paper with a Theorem statement and proof this confusing. Without additional evidence this is usually a sign of shallow analysis.

---

> > > ### Author Response · Authors · 2025-04-06
> > >
> > > Thank you very much for your thoughtful suggestion regarding the use of the notation $S$ in our paper. Your comment prompted us to carefully review the entire manuscript, and we have now clarified that $S$ in the proof is used exclusively as a diagonal matrix representing a collection of 1-sparse signals, and it does not conflict with any other usage of sparse codes (which are denoted by lowercase $s$) elsewhere in the paper. See the updated proof statement below:
> > >
> > > Let $K \geq 2$ and $P_K$ be a sparse distribution over $\mathbb{R}^N$, i.e., $\forall s \in \mathbb{R}^N: s \in \text{supp}(P_K) \iff \|s\|_0 \leq K$. This means that any sample has at most $K$ non-zero entries or, equivalently, the support of $P_K$ is a union over $K$-dimensional subspaces. The sources are linearly projected into an $M$-dimensional space, satisfying the restricted isometry property, where $K \log \frac{N}{K} \leq M < N$. A sparse autoencoder (SAE) with a linear-nonlinear (L-NL) encoder must have a non-zero amortisation gap.
> > >
> > > We appreciate your helpful feedback in clarifying these important details. Could you please confirm if this revision addresses your concerns fully, or if there are additional points you would like us to clarify further? Your suggestions have significantly improved our manuscript, and we greatly value your input.
> > >
> > > Thank you again for your valuable review.

---

### Decision · Program_Chairs · 2025-05-01

**Decision:**

Accept (poster)

**Comment:**

This paper uses compressed sensing theory to prove that sparse autoencoders (SAEs) are insufficient for accurate sparse inference. It also presents relevant experiments showing increased interpretability using more expressive encoders. Given the recent interest in SAEs, there is an interest in comparing them with classic approaches to sparse estimation.